Transcranial Doppler ultrasound to assess cerebrovascular reactivity: reliability, reproducibility and effect of posture

McDonnell Michelle N. 1 michelle.mcdonnell@unisa.edu.au
Berry Narelle M. 2
Cutting Mark A. 2
Keage Hannah A. 3
Buckley Jonathan D. 2
Howe Peter R.C. 2
1 School of Health Sciences, University of South Australia , Australia
2 Nutritional Physiology Research Centre, University of South Australia , Australia
3 School of Psychology, Social Work and Social Policy, University of South Australia , Australia
Jones Philip
Electronic publication date: 2013 Apr 9
Publication date: 2013
Volume: 1
Electronic Location ID: e65
Received 2013 Feb 19; Accepted 2013 Mar 20
Copyright: © 2013 McDonnell et al.
Copyright year: 2013
Copyright holder: McDonnell et al.
License: This is an open access article distributed under the terms of the Creative Commons Attribution License, which permits unrestricted use, distribution, and reproduction in any medium, provided the original author and source are credited.
License URL: https://creativecommons.org/licenses/by/3.0/

Keywords: Reliability, Reproducibility, Cerebrovascular reactivity, Posture, Doppler ultrasound

Funding: Dr. MN McDonnell and Dr. HA Keage are supported by Australian National Health and Medical Research Council Research Training Fellowships. The funders had no role in study design, data collection and analysis, decision to publish, or preparation of the manuscript.

==============================
Transcranial Doppler ultrasound (TCD) allows measurement of blood flow velocities in the intracranial vessels, and can be used to assess cerebral vasodilator responses to a hypercapnic stimulus. The reliability of this technique has not been established, nor is there agreement about whether the technique should be performed in sitting or lying postures. We tested the intra- and inter-rater reliability of measures of cerebrovascular reactivity (CVR) in 10 healthy adults, in sitting and lying postures. Participants underwent triplicate bilateral ultrasound assessment of flow velocities in the middle cerebral arteries whilst sitting and lying supine prior to and during inhalation of Carbogen (5% CO2, 95% O2) for 2 min. This procedure was performed twice by each of two raters for a total of four sessions. CVR was calculated as the difference between baseline and the peak blood flow velocity attained during CO2 inhalation. Intraclass correlation coefficients (ICCs) for intra-rater reliability were greater sitting than lying for both raters (e.g. Rater 1 ICC sitting = 0.822, lying = 0.734), and inter-rater reliability was also greater in sitting (e.g. sitting ICC = 0.504, lying = 0.081). These results suggest that assessment of CVR using TCD should be performed with participants sitting in order to maximise CVR measurement reliability.

Introduction

Transcranial Doppler (TCD) sonography allows real time measurement of blood flow velocity in intracranial arteries with high temporal resolution and has been used for over two decades to assess cerebral haemodynamics in healthy adults and in disease states. TCD can also be utilised to assess cerebrovascular reactivity (CVR), i.e. the vasodilation of cerebral arterioles in response to a physiological stimulus such as an increase in the partial pressure of arterial CO2 (hypercapnia) (Ainslie & Duffin, 2009; Willie et al., 2011). Several reports suggest that the cerebral vasodilator response to hypercapnia depends on the integrity of the vascular endothelium (Silvestrini et al., 2000; Krainik et al., 2005; Lavi et al., 2003) such that CO2 induced CVR could serve as a surrogate marker of endothelial function in cerebral arteries (Lavi et al., 2006). However, the reliability of the CVR method, particularly under differing postural conditions (i.e. lying vs sitting) has not been fully investigated.

Differing methods can be used to induce a cerebral arteriolar vasodilatory response, such as hypercapnia (using steady state breathing during 5% CO2 exposure or breath holding techniques) or medications such as acetazolamide (Dahl et al., 1995). Regardless of the stimulus, CVR is taken as the resultant change in mean blood flow velocity from a pre-stimulus baseline (Kernan et al., 2000; Willie et al., 2011). Mean blood flow velocity is typically measured in a conduit vessel such as the middle cerebral artery (MCA) which supplies a large proportion of brain tissue but will itself dilate minimally in response to the stimulus (Serrador et al., 2000). Thus changes of blood flow velocity in the MCA reflect changes in resistance to flow in the distal vasculature, i.e. arterioles in brain regions supplied by the MCA.

Repeatability of TCD measurements of MCA blood flow velocities was established nearly two decades ago. Demolis and colleagues confirmed that both intra-rater reproducibility and inter-rater reliability were satisfactory with strong correlations (r ≥ 0.9 in both cases) (Demolis, Chalon & Giudicelli, 1993). This study also established that mean velocities in the MCA vary with time of day, with significantly lower velocities recorded at 11am compared with 8am, and these variations in flow velocity follow a similar time course to diurnal variations in mean arterial pressure. This highlights the importance of conducting repeat experiments at the same time of day.

Reliability of CVR measures has previously been reported with the aim of comparing different methods of inducing hypercapnia. Intra-rater reliability of CVR measures was superior when induced by inhalation of 5% CO2 but this study did not address the reliability of different raters or different postures (Totaro et al., 1999). Others have compared measurements of CVR in lying and sitting postures and found that baseline and peak cerebral blood flow velocity (CBV) were lower in sitting compared to lying, but reproducibility of the method in sitting was not investigated (Mayberg et al., 1996).

This study aimed to determine the intra- and inter-rater reliability of measures of CVR in bilateral MCAs of healthy adult participants in sitting versus lying positions. Investigation of posture was performed due to the availability of ultrasound probes attached to a headband, allowing this technique to be performed sitting and lying supine. Sitting may be preferable for some participants, particularly the elderly who may find lying supine whilst holding a breathing tube and mouthpiece uncomfortable.

Methods

Participants

Ten adult participants (aged 20–64, mean age 33.8 ± 13.4 years [mean ± SD]) were recruited from staff and students of the host institution after giving their written informed consent. The study was approved by the institutional Human Research Ethics Committee. Participants were excluded if they had cardiovascular or renal disease, a history of diabetes, stroke and/or neurological conditions, were smokers or on nicotine replacement therapy, or were taking anticholinergic or psychotropic medications. Participants refrained from eating two hours prior to testing and caffeine four hours prior to testing.

Apparatus

Bilateral MCA blood flow velocities were measured with a Multigon TOC2M Neurovision TCD system (Multigon Industries Inc., New York, USA) using two 2-MHz transducer probes securely mounted on an adjustable headband. Time-averaged mean blood flow velocity measures (cm/s) were obtained every second (i.e. 1 Hz sampling rate) and stored for off-line analysis.

Procedure

Participants attended an air-conditioned laboratory on four separate occasions at the same time of day, with each visit separated by one week. Assessments in both the lying and sitting positions were conducted on two occasions by Rater 1 (MMcD) and on two occasions by Rater 2 (MAC); both raters had recently performed at least 25 assessments. The order of assessments, either sitting or lying supine, was randomised between raters and sessions. On arrival, the headband that holds the two 2.0 MHz transducers was placed over the head and adjusted for a comfortably firm fit. The 2-MHz transducers were then centred on the temple, adjusted until maximal amplitude signals were obtained bilaterally, at which time they were secured in the headband to ensure stability of the signals for the duration of the experiment. The precise positioning and recording of mean flow velocity in each MCA was performed as previously described (Aaslid, Markwalder & Nornes, 1982) through the trans-temporal windows, above the zygomatic arch. Participants breathed room air through a 2-way non-rebreathing valve and mouth piece with a nose-clip attached and were either seated comfortably in a chair with arm-rests, feet flat on the floor, or lying supine with one pillow for at least 10 min prior to commencing the testing. Participants were advised to perform “relaxed breathing, in and out through the mouth”.

After at least 10 min of breathing room air through the mouthpiece, measurement of CVR was performed during hypercapnia induced by breathing a mixture of 5% CO2–95% O2 (Carbogen) from a spirometer. This method was used in preference to breath holding or controlled hypoventilation to avoid concomitant hypoxia which can influence the ventilatory response (Ainslie & Duffin, 2009) and it has been shown to be more reproducible than other methods when performed by the one rater (Totaro et al., 1999). Beat-to-beat changes in mean CBV were measured bilaterally during a 30-s baseline period, then breathing the Carbogen mixture for 2 min. Each test was then followed by a 2 min rest period breathing room air again. This procedure was repeated three times both sitting and lying (counterbalanced) separated by a 10 min rest period after changing positions. Participants returned the following week at the same time of day to repeat the study with the same rater, then on two subsequent occasions with the other rater (counterbalanced).

Data analysis and statistics

The sample size of 10 participants was chosen in accordance with other reliability studies involving TCD (Harrer et al., 2011; Brodie et al., 2009). Recordings of bilateral CBV waveforms were averaged over the 30s baseline period in each of the triplicate assessments in each position (for each of the four separate testing sessions). The CBV waveforms obtained during the 2 min periods of hypercapnia were plotted against time and a spline curve was fitted (20% smoothing, LOESS filter) to aid determination of the peak blood flow velocity (Table Curve 2D, Systat Software Inc., CA, USA) (Fig. 1). Calculation of CVR involves comparing beat to beat changes in CBV to a pre-stimulus baseline (Willie et al., 2011; Wijnhoud, Koudstaal & Dippel, 2006) and is calculated: %CVR=100×(peak CBV−baseline CBV)/baseline CBV.

This was calculated separately for each of the three measurements in each position on each visit and the average of the three values was used for further analysis.

Figure 1 Application of curve fitting to CBV data.

Beat-to-beat cerebral blood flow velocities (CBV) obtained during the 2-min period of hypercapnia were plotted and fitted with a spline curve, enabling the peak of the curve to be identified as peak CBV.

All data were analysed using SPSS software (PASW Statistics 18.0) and graphs were created in Sigmaplot 12 (Systat Software, Inc.). Paired t-tests were used to verify differences between left and right sides and between sitting and lying CVR measures. Inter- and intra-rater reliability was established using intra-class correlation coefficients (ICC). As only two raters were used to score an average rating of CVR which is analysed in a two-way ANOVA model, this is evaluated using a Case 3, average rating or (3,1) ICC model, In accordance with Shrout & Fleiss (1979). Results were considered significant at P < 0.05.

Results

10 participants were enrolled in the study with 8 participants completing all four sessions. Two participants (authors MMcD and MAC) attended only two sessions, both with the same rater; thus intra-rater reliability data was obtained from 9 participants by each rater and inter-rater reliability was assessed for 8 participants.

Data were collected bilaterally at each session and values for left and right MCAs were compared. Depth of insonation was similar for left and right sides and between postures (e.g. 50.3 ± 4.1 mm in lying, 50.2 ± 4.1 mm in sitting). There were no differences between hemispheres when data from both testing sessions for both raters were combined (sitting P = 0.27, lying P = 0.46), allowing this data to be combined for further analysis (i.e. left and right MCA velocities were collapsed for all subsequent analyses).

Differences between lying and sitting

Individual CVR data from each test performed by both raters at both sessions were collapsed relative to sitting or lying. Differences in variability between the two positions were explored using Coefficients of Variation (CV) to quantify whether there was greater variation if measures were taken sitting rather than lying supine. CVs were calculated separately for sitting and lying measures obtained from each participant on each test occasion. There were no differences in variability of CVR measures obtained sitting versus lying (sitting average CV = 35.2 ± 6.6%; lying average CV = 36.7 ± 8.1%; paired t-test P = 0.5).

Intra-rater reliability

Measures obtained sitting and lying at two testing sessions by Rater 1 and Rater 2 were analysed using intraclass correlation coefficients to compare CVR measurements between sessions. Results for Rater 1 (Fig. 2) illustrate superior reproducibility between sessions when measured sitting (ICC(3,1) = 0.822; P < 0.001) compared with lying (ICC(3,1) = 0.734; P < 0.05).

Figure 2 Intra-rater reliability correlations in sitting and lying, Rater 1.

Correlations between %CVR measures taken on two occasions by Rater 1 in sitting (left, ICC (3,1) = 0.822) and lying (right, ICC(3,1) = 0.734). Simple linear regression lines are shown, using the least squares method (Sigmaplot 12).

Measures for Rater 2 were comparable, as shown in Fig. 3, with greater reproducibility in sitting (ICC(3,1) = 0.463; P = 0.12) compared with lying, although the correlations were non-significant (ICC(3,1) = 0.015; P = 0.49).

Figure 3 Intra-rater reliability correlations in sitting and lying, Rater 2.

Correlations between %CVR measures taken on two occasions by Rater 2 in sitting (left, ICC (3,1) = 0.463) and lying (right, ICC(3,1) = 0.015). Simple linear regression lines are shown, using the least squares method (Sigmaplot 12).

Inter-rater reliability

To demonstrate inter-rater reliability for both sitting and lying postures, data for each side (left and right) and each session (Session 1 and 2) were combined so that CVR values obtained by Rater 1 and Rater 2 could be compared as in Fig. 4. Intraclass correlation coefficients were determined and were significant for sitting (ICC(3,1) = 0.504; P = 0.028) but not lying (ICC(3,1) = 0.081; P = 0.41).

Figure 4 Inter-rater reliability comparisons showing superior reliability in sitting.

Correlations between %CVR measures obtained across both sessions by Rater 1 compared with Rater 2 for sitting (left, ICC(3,1) = 0.504) and lying (right, ICC(3,1) = 0.081). Simple linear regression lines are shown, using the least squares method (Sigmaplot 12).

Discussion

Assessment of CVR in healthy adults can be achieved with bilateral recording of CBV in the MCA with acceptable reproducibility and inter-rater reliability. This is the first study to demonstrate that measures of CVR obtained sitting were more reproducible than whilst lying. The reproducibility measures obtained were superior to those obtained by Totaro and colleagues (Totaro et al., 1999) (ICC = 0.43 when measures were repeated by a single investigator after 24 h). Reproducibility measures for both raters were greater in sitting compared with lying (ICC(3,1) = 0.822 and 0.463 in sitting respectively for both raters) which shows a strong agreement between sessions. Intra-rater reliability was greater for the more recently experienced sonographer, Rater 1, confirming previous reports (Martin, Thomas & Caron, 1993; Shen et al., 1999). Inter-rater reliability was only fair, indicating that the same rater should be used where possible for measurements in research trials and clinical settings. It is important to note that these values for reliability and reproducibility are valid for measurement of hypercapnia at 5% CO2 and 95% O2, providing a strong vasodilatory stimulus but we acknowledge that for near maximal vasodilation to occur, 10% CO2 may be required (Goode et al., 2009).

We have recently shown (Wong et al., 2013) that regular dietary supplementation with a wild green oat extract can enhance systemic endothelial function assessed by the technique flow mediated dilatation of the brachial artery and cerebral endothelial function assessed in the MCA by CVR to hypercapnia. Even though these responses were of similar magnitude (∼40% increase in both cases) they were not correlated within individuals, indicating the likely independence of underlying mechanisms and pathophysiological changes in endothelial function between systemic and cerebral arteries. Thus, while the technique of flow mediated dilatation of the brachial artery is well established, TCD has the potential to determine the efficacy of therapeutic treatments to improve vascular health. The present study provides evidence of good intra-rater reliability, particularly with an experienced sonographer, but only acceptable inter-rater reliability using Carbogen as the hypercapnic stimulus to measure changes in CVR within-individuals in response to therapeutic interventions. The results of this study suggest that performing the test in sitting is preferable to lying to assess changes in cerebral endothelial vasodilator function which may be appropriate for elderly or disabled populations who may be involved in research investigating dietary or behavioural interventions to improve cerebrovascular function.

The technique of TCD to assess changes in cerebral blood flow velocity, although widely used, does have a known limitation, in that we cannot prove that the diameter of the MCA does not change during the period of hypercapnia. The method used here and by others assumes that the MCA diameter was constant during hypercapnia (Serrador et al., 2000). A number of studies have demonstrated that the diameter of the MCA does not change significantly during craniotomy (Giller et al., 1993), after the release of thigh cuffs to assess cerebral autoregulation (Newell et al., 1994) or a similar TCD technique using the Doppler power signal to measure MCA diameter (Poulin, Liang & Robbins, 1996). However, there have been reports in animal and human studies that small changes in vessel diameter are possible, particularly during hypocapnia (Du Boulay & Symon, 1971; Du Boulay et al., 1972).

There are several limitations to our study. We only used two raters and additional raters may provide more accurate measures of ICCs. Further, we did not measure blood pressure of these healthy individuals, nor did we relate our measurement of CVR to end-tidal CO2 by performing gas analysis on expired air during the experiments. Our results, therefore, are limited to relative changes in CBV in the MCA in response to 5% CO2, rather than relative increase in CBV per kiloPascal rise in end-tidal CO2 (Wijnhoud, Koudstaal & Dippel, 2006). The important metric to record when assessing CVR is the beat-to-beat change in CBV from a pre-stimulus baseline (Willie et al., 2011) and the method we have used for this and other clinical studies satisfies this requirement and avoids the subject discomfort which can occur with administration of ≥ 8% CO2 (Ainslie & Duffin, 2009). This method has been used by others to assess cerebrovascular endothelial function (Wijnhoud, Koudstaal & Dippel, 2006; Schwertfeger et al., 2006) and has been shown to be preferable to normalizing to absolute increase in end-tidal CO2 in predicting stroke and TIA risk in patients with carotid artery stenosis and occlusion (Markus & Cullinane, 2001).

The data from this study indicates that the assessment of CVR using TCD provides more reliable values when performed sitting compared with lying, particularly with a more experienced sonographer. Future studies should be performed with participants in the sitting position rather than lying to maximise CVR responses and to optimise inter- and intra-rater reliability.

Additional Information and Declarations

Competing Interests

Author Contributions

Human Ethics

We have no competing interests to disclose.

Michelle N. McDonnell conceived and designed the experiments, performed the experiments, analyzed the data, contributed reagents/materials/analysis tools, wrote the paper.

Narelle M. Berry conceived and designed the experiments, performed the experiments, contributed reagents/materials/analysis tools, wrote the paper.

Mark A. Cutting performed the experiments, analyzed the data.

Hannah A. Keage contributed reagents/materials/analysis tools, wrote the paper.

Jonathan D. Buckley conceived and designed the experiments, wrote the paper.

Peter R.C. Howe conceived and designed the experiments, contributed reagents/materials/analysis tools, wrote the paper.

The following information was supplied relating to ethical approvals (i.e. approving body and any reference numbers):

University of South Australia Human Research Ethics Committee.

Approval for this study, application number 0000022568 was granted in June 2011.

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
