# Peer review of "Transcranial Doppler ultrasound to assess cerebrovascular reactivity: reliability, reproducibility and effect of posture"

_PeerJ, doi:10.7717/peerj.65_

## Round 0.1 · original submission · Minor Revisions

Dear Dr McDonnell and colleagues,

Thank you for submitting this interesting article to PeerJ.

Overall, this is an important paper highlighting both the strengths and weaknesses of transcranial Doppler to measure cerebrovascular reactivity in healthy volunteers. I would highlight the important points made by the Reviewers, especially the following:
- please comment on how the sample size was determined
- please ensure that the Introduction relates specifically to the investigation performed (ties to other research not directly relevant should be made *succinctly* in the Discussion section)
- please clarify why the Abstract contains ICCs which are different than those in the manuscript proper
- please explain why the number of data points in Figures 3 and 4 are so dramatically different.
- for Reviewer 2, please disregard the comment made about "temple"


I would also personally make the following comments.

Specifics
* * *
- line 14 states that acetazolamide can be used to induce *hypercapnia*, but acetazolamide causes *hypocapnia* in healthy volunteers. (See Acetazolamide and Breathing - Does a Clinical Dose Alter Peripheral and Central CO2 Sensitivity? http://ajrccm.atsjournals.org/content/160/5/1592.long - let me know if I have misunderstood what you are trying to say here.)
- line 57: specify what is meant by "mood medication". Do you mean antidepressants, mood stabilizers, anti-psychotics, or all three? It may be easier to state "psychotropic medications".
- line 80: add a comma between "mouthpiece" and "measurement"
- line 83: do you mean "hypoventilation" (increased PaCO2) rather than "hyperventilation"?
- discuss the effect of changing both the FiO2 AND the FiCO2 when the participants were inhaling the Carbogen mixture (rather than just changing the FiCO2 on its own). Does the increase in FiO2 from 0.21 to 0.95 have any impact on the interpretation of the results?
- a short sentence in the Methods section, using the seminal paper by Shrout and Fleiss as a guide, explaining which ICC model was used (i.e. what ICC (3,1) actually means), would be useful for most readers of this paper. (Reference: Psychological Bulletin, 1979, Vol. 86 No. 2, pp. 420-28)
- line 132: report this P value as either "P<0.001" or "P<0.0005"
- line 163: insert a comma after "velocity"AND after "used"
- line 177: kilopascal should be one word
- line 183: add "of" after "administration"
- line 189: add "-" after "inter"
- line 190: change "sitting" to "the sitting position"
- state which software was used for construction of the figures, if different from the main statistical analysis. To my eye, it appears like GraphPad Prism might have been used. This is relevant since there was a regression line plotted.
- Figures 3, 4, and 5: I'm assuming the straight lines are ordinary least-squares regression lines, but please make what these lines are explicit in the Figures' captions.

More Broadly
* * *
- isn't one of the main points of this manuscript the fact that the inter-rater reliability seems to be quite poor (in either position)? This is concerning for any diagnostic method. Discuss this as a limitation of the technique, especially if TCD is to be used more widely in the future.
- future research should ideally study many more raters to obtain a more precise estimate of the ICCs. Conclusions based on this small study may be erroneous and biased due to the small number of raters. This limitation should be highlighted.

Please ensure you respond to each of the points made by the Reviewers as well as the additional points above, and thank you again for submitting your work to PeerJ!

Reviewer 1 ·

Basic reporting

The aim of this study was to assess inter- and intra-rater reproducibility of TCD measures. The question is of interest and relevant to research and clinical studies. The manuscript is well written and easy to read.

Experimental design

How did you decide on 10 participants?

Validity of the findings

no comment

Additional comments

Introduction:
Only the last sentence of the first paragraph is really relevant to this study. Indeed much of the first 3 paragraphs could be deleted. The underlying assumption for TCD are well known and could be alluded to in the Discussion. Dietary supplements, endothelial function etc seem irrelevant unless they are known to have a significant postural influence. This study posed a simple question – what is the inter- and intra-rater reproducibility. The Introduction simply needs to present a case for these being posturally influenced.

Methods:
How did you decide on 10 participants?

Discussion:
Lines 156-162 can be deleted.
Lines 177 -186 can be significantly shortened. The author’s research interests are not what this study was about.

Fig 1 could be deleted.

Reviewer 2 ·

Basic reporting

This is an interesting manuscript.
The introduction demonstrates appropriately how the work fits into the field of knowledge.
Some sentences would benefit some clarification: lines 27 and 28, lines 167, 168 and 169
Line 71 word ‘’temple’’ – not in the dictionary
Line 144 – recording
Line 169 – should the second word (to) be removed?
Line 171 - hypocapnia

Experimental design

From my understanding of the study, the primary objective of the study is to determine the intra and inter-rater reliability of measures of CVR in bilateral MCA using TCD in sitting versus lying positions. Could you please be more specific in describing the objective in the manuscript (line 44-49).
How was calculated the sample size? What is the power of this study? Please add this information in the section ‘’Data analysis and statistics’’.

Validity of the findings

Intra-rater reliability: Although ICC for rater 1 in the sitting and lying position can be interpreted as almost perfect agreement and strong agreement, ICC for rater 2 can only be interpreted as fair agreement and poor agreement. Could you please discuss that point when interpreting the results?
Since the ICC is poor to fair for rater 2 measurements, how can this affect the results of the ICC calculation regarding inter-rater reliability?
Please explain why the numbers of data (number of dots) are different in figure 3 and 4.
The ICC presented in the abstract are labelled as ‘’Rater 1’’ but are different than those in the manuscript. Please clarify.
The last sentence should be part of the discussion instead of being the conclusion.

Additional comments

No comments

---

## Round 0.2 · Minor Revisions

Dear Dr McDonnell and colleagues,

Thank you for submitting an excellent revision which incorporates the changes and suggestions made by the Reviewers.

However, there are still a few very minor issues to be dealt with before publication, all of which will take you only a brief time to correct:

- in the title and on line 168, "Doppler" (proper name) should be capitalized. (My apologies for missing this the first time around.)
- the sentence (lines 13-15) "Differing methods can be used to induce hypercapnia, including steady state breathing during 5% CO2 exposure, breath-holding techniques or medications such as acetazolamide can be used to induce a strong vasodilatory response (Dahl et al., 1995)." no longer makes sense. Do you mean something like "Differing methods can be used to induce a cerebral arteriolar vasodilatory response, such as hypercapnia (using steady state breathing during 5% CO2 exposure or breath-holding techniques) or medications such as acetazolamide (Dahl et al., 1995)." Please amend as you see fit.
- although "cerebral blood flow velocity" is defined on lines 76-77, it should actually be defined at the time of its first usage, on line 33
- line 72: insert a space between "95%" and "O2"
- line 88: "Lowess" should either be all uppercase (as it stands for "LOcal regrESSion") or all lowercase
- line 162: please insert a comma after "velocity" to make this sentence read better
- line 175: please insert a comma after "therefore"
- you refer to "beat-to-beat" differently in the manuscript. Please unify these to "beat-to-beat" (two dashes) in lines 76, 178, 271, and Figure 1's caption (above the figure itself)
- there is no need to capitalize "Spline" (for Figure 1, both on page 14 and above the Figure itself)
- the Figure caption for Figure 2 is incorrect (it is the one for Rater 2, not Rater 1). Note that the captions are correct on Page 14 of the manuscript, it is only the caption above the Figure itself which is incorrect.
- I realize you want to emphasize the clinical import of these findings, but there is still a problem with the last few lines of the Conclusions (lines 188-191), as statements are made which weren't tested in this trial. Specifically, no elderly or disabled participants were studied, and no dietary or behavioural interventions were studied. It is inappropriate to comment on these populations and interventions in the final lines of this study report. My suggestion would be to move this concept (which is important and deserves mention) to the Discussion, where the authors' interpretations and speculations would be more appropriate. The Conclusions should be entirely based on what was actually studied in the trial.

Thank you again for submitting an excellent revision.

---

## Round 0.3 · accepted · Accept

Dear Dr McDonnell and colleagues,

Thank you for dealing with the outstanding issues identified and for re-submitting your manuscript in such a timely fashion.

I am pleased to now accept your manuscript for publication in PeerJ. I look forward to seeing more of your work in the future!